# Untangling Depression in Schizophrenia: The Role of Disorganized and Obsessive-Compulsive Symptoms and the Duration of Untreated Psychosis

**DOI:** 10.3390/biomedicines12112646

**Published:** 2024-11-20

**Authors:** Georgi Panov, Silvana Dyulgerova, Presyana Panova, Sonia Stefanova

**Affiliations:** 1Psychiatric Clinic, University Hospital for Active Treatment “Prof. Dr. Stoyan Kirkovich”, Trakia University, 6000 Stara Zagora, Bulgaria; 2Medical Faculty, University “Prof. Dr. Asen Zlatarov”, 8000 Burgas, Bulgaria; 3Medical Faculty, Trakia University, 6000 Stara Zagora, Bulgaria

**Keywords:** schizophrenia, psychosis, positive, negative, disorganized symptoms, obsessive-compulsive symptoms, duration of untreated psychosis, depressive symptoms, depression, biological antagonism

## Abstract

**Background:** Schizophrenia is a complex disorder characterized by positive symptoms (e.g., hallucinations), negative symptoms (e.g., social withdrawal), and disorganized symptoms (e.g., thought disorder). Alongside these, cognitive and depressive symptoms often emerge, with depressive symptoms sometimes dominating the clinical picture. Understanding the factors that influence the development of depressive symptoms in schizophrenia could clarify the dynamics between depressive and psychotic symptoms and guide clinical interventions. **Methods:** A total of 105 patients with schizophrenia (66 women, 39 men) were assessed using several clinical scales: PANSS, BPRS, DOCS, DES, HAM-D, and the Luria-Nebraska Neuropsychological Battery for cognitive evaluation. Statistical analyses, including correlation and regression, were conducted using SPSS to determine the significance of associations. **Results:** Disorganized and obsessive-compulsive symptoms were identified as primary factors associated with depressive symptoms in patients with schizophrenia. Conversely, a longer duration of untreated psychosis was linked to a lower severity of depressive symptoms, suggesting that early intervention may alter the depressive symptom trajectory. **Conclusions:** Here, we suggest a complex interaction between psychotic and depressive symptoms, possibly indicating a biological antagonism. The association of depressive symptoms with disorganized and obsessive-compulsive features may reflect an adaptive psychological response, attempting to stabilize amidst the disintegration of schizophrenia. These insights support a more integrated approach to treatment, addressing both psychotic and depressive symptoms to improve patient outcomes.

## 1. Introduction

Schizophrenia is a complex nosological entity that results from the interaction of genetic and epigenetic factors [1,2,3,4]. Genetic predisposition is easily detectable in the analysis of family history. This encumbrance in an etiological aspect cannot be accepted too categorically due to the influence of the impact of learned patterns of behavior, as well as the lack of stable parental role models. These factors can also be considered epigenetic impacts [5]. Analysis of the interaction of epigenetic influence and genetic polymorphism shows a relationship with both the development of the schizophrenia process and the effect of antipsychotic therapy [6]. The clinical picture of schizophrenic disorders is diverse, but positive, negative, and cognitive symptoms have been analyzed as the main storyline. In patients with schizophrenia, many other symptoms are also superimposed, such as anxiety, affective, and obsessive-compulsive disorders, as well as disorders in the perception of time and the perception of personal space [7,8]. All these deviations in mental status are also associated with changes in both metabolism and immunological markers [9,10]. All this multiplicity, both in the etiological and clinical aspects, poses the question of searching for a complex approach and therapeutic behavior that is consistent with the main and priority aspects in the clinical presentation [11,12,13].

Depressive symptoms and psychotic symptoms have many similarities and differences. These similarities and differences can be observed at the structural, functional, neurotransmitter, and metabolic levels. In both conditions, the analysis of structural disorders shows that we have a loss of gray brain matter in different parts of the brain, with a corresponding predilection for changes in them. In depression, changes are more pronounced in the hippocampus, prefrontal cortex [14], and the amygdala and white brain matter [15]. In schizophrenia, they are associated with gray matter loss and cortical thickness reduction [16], as well as hippocampal and temporal lobe changes [17]. Functional changes in depression are associated with hyperactivity in the default mode network (DMN) [18] and hypoactivity in the executive control network (ECN), as well as hyperactivation of the amygdala and anterior cingulate cortex (ACC) [19]. In psychosis, DMN hypoactivity and dysfunction [20] and dysregulation in the salience network (SN) [21] altered thalamocortical connectivity and dopamine system dysfunction [22]. Differences in the neurotransmitter system are also observed in both conditions. In depression, there are major changes in serotonin and norepinephrine [23], as well as in the glutamate and hypothalamic-pituitary-adrenocortical (HPA) axis [24]. In schizophrenia, the main neurotransmitter changes are found in dopamine and glutamate [25,26], as well as in the dysfunction of NMDA receptor activity [27]. Differences can be found in a purely metabolic aspect, related to the metabolism of kynurenic acid. Schizophrenia shows a KYNA elevation that interferes with receptor activity crucial to cognitive function and psychosis [28] while depression often shows a neurotoxic shift, increasing quinolinic acid [29]. These changes also warrant the search for therapeutic strategies to influence kynurenic acid metabolism [30,31,32].

Depressive symptoms are frequent and characteristic symptoms observed in patients with schizophrenia. On the one hand, they can be observed as clinically established symptoms in the post-psychotic period, reaching up to 50% in patients with first episodes and up to 25% in those with subsequent episodes. The presence of these episodes is associated, according to some authors, with a poor prognosis [33,34]. The opinion of other authors takes an opposite position, associating affective symptoms with a favorable course of schizophrenia [35,36,37], especially in those with high values on the depression scale [38] and in those with high values on the mania scale [39]. The importance of depressive symptoms in the development of the schizophrenic process has led to their special classification in ICD 10 and ICD 11 as post-schizophrenic depression [40,41]. These divergent data indicate that depressive symptoms themselves are most likely to have a diverse genesis, which in turn is associated with a different prognosis. On the other hand, apart from being a clinically evident phenomenon, depressive symptomatology can be observed as a subclinical persistent finding in the clinical picture of patients, not always demonstrably presented and mixed with the main symptomatology. A study conducted with the aim of analyzing depressive symptoms in patients with schizophrenia showed that they are moderately pronounced and persistent over time in patients with resistance to treatment and are relatively mild in those in remission [42]. These data give us reason to conclude that smoldering depressive symptoms are associated with resistance to the schizophrenic process, while the appearance of pronounced depressive and manic symptoms is associated with a good prognosis.

In the analysis of the factors related to the presence of depressive symptoms, the author team established a relationship between the negative subscale of the PANSS and the scale of general psychopathology with sociodemographic factors such as loneliness, isolation, and lack of support [43,44]. On the other hand, whether there is a connection with sociodemographic factors as a cause of depressive symptoms or whether sociodemographic factors are a consequence of the development of the schizophrenic process with the gradient loss of social connectivity associated with its progression is a question that can be discussed. In this case, it is difficult to give an exact answer, as an analogy can be made with the question: "Which came first - the chicken or the egg?"”

There are multivariate analyses that present a different clinical value of depressive symptoms in patients with schizophrenia. This lack of consistency in research gave us reasons to look for and establish the relationship between depressive symptoms and other factors and symptoms in patients with schizophrenia.

In the analysis of the factors related to depressive symptoms, we considered it appropriate not to include the sociodemographic data. Due to the fact that they also appear as a result of the persistence of the main symptoms of the schizophrenia process, depressive complaints, in themselves and as the development of social maladaptation, appear to be directly deducible from the schizophrenic process.

## 2. Materials and Methods

We conducted a study of 105 patients with schizophrenia, of whom 66 were women and 39 were men. The examination and evaluation of the patients was carried out in the Psychiatric Clinic of the University General Hospital for Active Treatment —Stara Zagora. Patients were admitted for observation and treatment after consecutive psychotic episodes. The initial examination of the patients was performed in an outpatient setting, and after giving informed consent, they were admitted for treatment and assessment of the condition. Patients were recruited and followed for the period from 2017 to 2022.

All patients undertook the Hamilton depression and anxiety scales, the dissociation scale, the obsessive-compulsive symptom analysis scale (DOCS), and a memory assessment, using the Luria test [45,46,47,48,49].

The initial analysis and follow-up were performed in an inpatient setting, and later, the observation and follow-up of their condition were performed in an outpatient setting.

Patient inclusion criteria:A series of psychotic episodes;Evaluation of symptoms according to the PANSS and BPRS scales [50,51]Prospective observation for at least 12 weeks;Administration of at least two trials of antipsychotic drugs at a dose equivalent to or greater than 600 mg chlorpromazine equivalents to achieve remission and assess resistance.

Exclusion criteria:Mental retardation;Abuse of psychoactive substances;Presence of organic brain damage;Accompanying progressive neurological or severe somatic diseases;Marked personality change (according to the DSM 5 and ICD 10 and ICD 11 diagnostic tools) [40,41,52,53];First psychotic episode.

### Statistical Analyses

For the purpose of the research, the capabilities of the statistical package SPSS version 26 were used. The methods used were tailored to the specifics and objectives of the research. Correlation analysis and regression analysis were used. Regression analysis was conducted, with the dependent variables being depressive complaints assessed with the Hamilton scale. Independent variables were age, disease onset, duration of untreated symptoms, duration of schizophrenia process, body mass index, PANSS positive, negative, and disorganized symptoms, dissociation scale, and obsessive-compulsive symptom scale. An assessment of the co-linearity index was also made to measure the significance and reliability of the obtained results. We have also used analysis of variance (ANOVA) as a statistical formula used to compare variances across the means (or average) of different variables. A correlation analysis was also conducted to find a relationship between the analyzed variables. On the other hand, correlation analysis and its results are a prerequisite for the choice of method when conducting regression analysis.

The same group of patients was studied in order to investigate other clinical indicators such as obsessive-compulsive, dissociative, and cognitive symptoms, lateralization of brain processes, the effect of the administration of the first antipsychotic drug, and the role of gender in patients with schizophrenia [42,54,55,56,57,58,59,60]. All patients gave written informed consent before being admitted to the clinical facilities and undergoing diagnostic tests and therapy.

The study was conducted in accordance with the Declaration of Helsinki and was approved by the Ethics Committee of the University Hospital “Prof. Dr. Stoyan Kirkovich” Stara Zagora, protocol code TR3-02-242/30 December 2021.

## 3. Results

### 3.1. Descriptive Statistics of the Sample

Out of a total of 105 patients with schizophrenia, 66 were female and 39 were male. The mean age of patients was 37.13 years (Table 1). The minimum age was 21 years and the maximum was 62 years.

In males, we found that the average level of depression was 11.44, and the median was the same as in females, which was 12, with a standard deviation of 4.179.

We found a slightly higher level of depressive symptoms in females, but without statistical dependence on the level of depressive scores between the sexes.

### 3.2. Correlations Between Clinical Scales

We conducted a correlation analysis in order to assess the strength of the relationship between the scales we analyzed: the depression scale, the PANSS negative, positive, and disorganized scales, the dissociation scale, and the OCS scale, as well as short-term memory fixation (Table 2, Table 3 and Table 4).

This analysis reveals that depressive symptoms are modestly associated with longer illness duration, suggesting that the chronicity of schizophrenia may contribute to increased depressive symptoms over time. However, onset age and duration of untreated symptoms do not appear to have direct impacts on depression levels. These findings point to a potentially cumulative effect of long-term schizophrenia on depressive experiences, which could suggest treatment approaches that address the emotional impact of prolonged illness.

The analysis of data in Table 3 reveals that depressive symptoms are moderately associated with overall psychiatric severity (PANSS and BPRS), particularly with disorganized symptoms. The findings suggest that disorganization, along with positive symptoms, may play a more significant role in the experience of depressive symptoms in schizophrenia than negative symptoms do. These insights highlight the importance of addressing both mood and cognitive disorganization in managing depressive symptoms within this population.

The analysis reveals that depressive symptoms are strongly associated with anxiety and obsessive-compulsive symptoms. In contrast, there is little to no significant relationship between depressive symptoms and dissociation or fixation. The strong link between anxiety and depression highlights the importance of addressing both conditions in clinical settings, as they often exacerbate each other. The associations found between dissociation and obsessive-compulsive symptoms suggest that these constructs may warrant further exploration, particularly in their relationship with depressive experiences. Also striking is the negative correlation between depressive symptoms and short-term memory reflected by fixation in this analysis. Overall, we can say that these findings emphasize the interconnected nature of various psychiatric symptoms and the need for comprehensive assessment and treatment strategies.

In order to establish the most important factors influencing the appearance of depressive symptoms in patients with schizophrenia, we conducted a regression analysis. Regression analysis was conducted, with the dependent variables being depressive symptoms. Independent variables were age, age of onset, duration of untreated symptoms, duration of schizophrenia process, body mass index (BMI), PANSS positive, negative, and disorganized symptoms, dissociation scale, and obsessive-compulsive symptom scale. The results are presented in Table 5 and Table 6.

Regression analysis identified three main predictors and models.

These predictors are:PANSS Disorganized Symptoms: The strong positive impact of disorganized symptoms on depression highlights the significant role of cognitive and perceptual disorganization in contributing to depressive symptoms. This may reflect the distress and impairment that disorganized thinking and behavior impose, potentially fostering feelings of depressive complaints.Duration of Untreated Symptoms: The negative relationship here suggests that individuals who experienced shorter periods of untreated psychosis report higher depressive symptoms. This finding could imply that early intervention mitigates long-term depressive symptoms, or that those who seek help sooner may be more likely to experience depression due to increased insight of their symptoms.Obsessive-Compulsive Symptoms (OCS): The positive correlation between OCS scores and depressive symptoms implies that the presence of obsessive-compulsive symptoms may aggravate depressive symptoms, possibly due to the distress and mental strain associated with them.

## 4. Discussion

Our study showed that gender distribution is not a factor related to the appearance of depressive symptoms in patients with schizophrenia. In the main population, there is evidence that there is a clear distinction between the sexes, with depressive symptoms being more characteristic of females. This is also directly deducible from the differences between the sexes, related to biological, psychological, and social factors, which are a prerequisite for the prevalence of depressive symptoms in persons of the female gender [61]. This is not the case in patients with schizophrenia. One study examines gender differences in the onset and progression of schizophrenia, including the expression of depressive symptoms, and suggests that typical gender differences in depression may not apply to individuals with schizophrenia [62]. Our results support this study by showing that schizophrenia patients did not differ in terms of the expression of depressive complaints. Other articles have also discussed how schizophrenia may diminish the usual gender differences seen in mood disorders, with a focus on the expression of depressive symptoms [63,64,65]. Other studies have attempted to analyze the reasons for the lack of differences in the occurrence of depressive symptoms in patients with schizophrenia. They examine and develop the concept of blunted gender differences in schizophrenia [66].

Another transcultural study also shows that there is a reduction of gender differences as a consequence of the influence of the schizophrenic process [67]. We also registered these blurred boundaries of gender differences in schizophrenia when conducting an assessment of the distribution of gender roles in patients with schizophrenia [57]. Our data also support the idea of blunted gender differences. The results of these studies give some authors reason to consider the idea of a strictly individual view and approach to these patients, not one based only on gender differences [68,69].

We find a directly proportional relationship between the expression of disorganized symptoms in schizophrenia and depressive symptoms. This dependence can be considered in the context that disorganized symptoms, unlike positive and negative symptoms, are directly related to functional impairment, which in turn leads to depressive complaints [70]. Negative symptoms are not perceived as ego-dystonic in the sense of suffering, and thus they do not lead to depressive complaints, although an overlap between depressive and negative symptoms may be observed outwardly, further impairing social functioning [71]. This close relationship between disorganized and depressive symptoms, as was shown in our previous study between disorganized and obsessive-compulsive symptoms [54], gives us reason to consider the idea of symptom formation as a consequence of the development of a process of “disintegration” in patients with schizophrenia. We can view depressive, positive, and obsessive-compulsive symptoms as defense mechanisms against the “disintegration” underlying the schizophrenic process [72,73,74].

This study also gives us an explanation of why we find obsessive-compulsive symptoms as a factor that is, to a large extent, a determinant of the appearance of depressive symptoms. Obsessive-compulsive symptoms themselves lead to severe stress and can further trigger the onset of depressive symptoms. Obsessive-compulsive symptoms in schizophrenia can also be seen as an adaptive phenomenon, a reaction against the chaos of disorganized symptoms [75].

We find the greatest association of depressive symptoms with disorganized and obsessive-compulsive symptoms. Another study of ours, which analyzed obsessive-compulsive symptoms in patients with schizophrenia, showed that they were also highly associated with the presence of disorganized symptoms [54]. Is it precisely the disorganized symptoms, symptoms without organization, that do not seek their secondary organization as psychotic, obsessive-compulsive, and depressive symptoms? Other studies also find a link between disorganized and depressive symptoms [76,77,78,79,80,81]. When conducting a correlation analysis, we found a high correlation dependence of depressive complaints, both with the individual subscales of the PANSS and with the BPRS scale. We found a higher correlation significance of depressive symptoms with disorganized symptoms and with the BPRS scale. The explanation of this observation can be found in the research that conducted a comparison between the PANSS and BPRS scales. The PANSS scale is much more specific than the BPRS. The BPRS scale reflects more the main psychopathology, which is also related to an overlap with some depressive and disorganized symptoms, which is the reason for their higher correlation coefficient [82,83,84].

Our study found that the duration of untreated symptoms was inversely related to the development of depressive symptoms. We find that the longer the duration of untreated psychosis, the lower the likelihood of developing depressive complaints. How should we analyze this observation?

In the context that psychosis is the primary and main disease, depressive symptomatology appears as an additional symptom in the course of the evolution of the disease or, on the other hand, as a side effect and as a symptom as a result of therapy. These relationships can be considered in the context of the relationship between dopamine blockade and depression, cognitive impairment associated with dopamine blockade, fatigue, sedation, motor side effects, etc. [78,85,86] . This is also the classic approach to looking at these relationships, and for this reason, in the F20 rubric of ICD 10 and 11, we have a sub-section on post-schizophrenic depression. Our data on the inverse correlation between depressive symptoms and duration of untreated psychosis prompts us to discuss another hypothesis. Since depressive symptomatology appears, on the one hand, as one of the main prodromal symptoms in schizophrenia, on the other hand, it represents a symptom that we can follow in the course of the schizophrenic process. From this point of view, the interrelationships between depressive and psychotic symptoms give us reason to ask the questionCan we not take a mirror look at these processes?

Depression is the primary disorder in which, in some cases, psychotic symptomatology is superimposed, which is clinically significant, expressed, and meets the diagnostic criteria of the main classification systems and, as such, requires antipsychotic therapy. In support of this observation comes the fact that often-expressed depression has psychotic symptoms that require antipsychotic therapy in parallel with the use of antidepressants [87,88,89,90,91].

On the other hand, the use of antidepressants has an effect on patients with schizophrenic disorder, even in the absence of clinically expressed depressive symptoms [92,93,94,95].

We can also judge the proximity of the two states by their prognosis. Both schizophrenia and psychotic depression have a similar prognosis, despite the presence of certain differences [96,97,98,99], which, again, gives us reason to question what is the primary underlying disorder. On the other hand, perhaps we should just accept that the question is largely rhetorical and that the two states are interwoven as one common entity.

These clinical observations give us reason to look at the biochemical disturbances in order to find similarities between them. In both conditions, a biochemical imbalance is established. In both conditions, there is a dopamine imbalance, disorders of serotonin, glutamate transmission, GABA mediation, an increased level of inflammatory factors, and a generally increased inflammatory background, associated with a change in cortisol secretion [100,101,102,103].

If we look at DMN changes in depression and psychosis, we find that they are opposite. While, in depression, we have hyperactivity [18], in schizophrenia, we have altered activity related to the loss of connectivity or altered activity to hypoactivity [20]. This process is most likely also related to the disorganization in the sense of “I”, which provokes the creation of clinical symptoms such as positive or depressive or OCD [104]. How can we explain, from this perspective based on DMN dysfunction, the inverse relationship between the duration of untreated psychosis and depressive symptoms? The prolonged state of dysfunction of this neuronal network over time is related to strengthening the loss of connectivity at the functional and neuronal level, where it becomes even more difficult to react with “hyperfunction” of the same system and transition to depression.

In patients with resistant schizophrenia, depressive symptoms are registered in the range of moderately expressed [42]. These results can be commented on in light of our observation that schizophrenic disorder, which is essentially a form of disorganization of the psyche, finds its clinical expression in the construction of disorganized symptoms as a clinical phenomenon. The organism is not able to reach a new form of mental allostasis, regardless of whether it is a psychotic or depressive episode, and remains in an unbalanced state with persistent disorganized symptoms, which also show the highest degree of resistance in individual studies [105,106,107].

Limitation: The limitation of our study is related, on the one hand, to the number of observed patients. In order to answer fundamental questions related to the perception and analysis of the clinical dynamics of larger nosological categories, observation of a large number of patients with a long follow-up period is necessary, which we are currently unable to provide. The observation of these patients continues, which gives us the opportunity to follow the pathoplasticity of the described phenomenology, which we hope to be able to present in the future.

Another limitation of our study is that it is purely clinical, as we did not have the opportunity to analyze the functional connections in individual neuronal networks, metabolic disorders, and individual registered inflammatory markers characterizing these processes.

## 5. Conclusions

We found that the presence of disorganized and obsessive-compulsive symptoms is associated with the appearance of depressive symptoms in patients with schizophrenia. Additionally, an inverse relationship between the duration of untreated psychosis and the onset of depressive symptoms suggests that early intervention may impact the development of depressive features. These findings offer valuable insights for clinical practice, potentially enabling the prediction of individual symptom dynamics throughout the course of the schizophrenic process. Moreover, these observations prompt a reconsideration of the relationship between depression and psychosis as conditions that often exhibit convergent aspects in therapy. Rather than viewing them as separate entities, our results align with the concept of a single underlying pathology—a unified disease process that may express itself through varying symptom profiles over time. This perspective encourages a holistic approach to treatment, addressing core vulnerabilities that underpin both psychotic and depressive symptoms in a continuum of mental health disorders.

## Figures and Tables

**Table 1 biomedicines-12-02646-t001:** Some of the main characteristics of the group of patients.

Age (years)	37.13
Age of onset of SZ (years)	25.51
Duration of untreated period/months/	14.78
Duration of SZ (years)	11.77
BMI	26.9562
Height (cm)	168.55
Sex (M/F)	39/66

**Table 2 biomedicines-12-02646-t002:** Degree of statistical significance between the depression scale and the onset of the illness, duration of the illness, and duration of untreated symptoms in patients with schizophrenia.

		Onset of the Illness	Duration of Untreated Symptoms	Duration of Sch	Hamilton D
Onset of the illness	Pearson Correlation	1	−0.350 **	−0.313 **	−0.039
	Sig. (2-tailed)		0.000	0.001	0.694
Duration of the untreated symptoms	Pearson Correlation	−0.350 **	1	0.148	−0.104
	Sig. (2-tailed)	0.000		0.133	0.292
Duration of the illness	Pearson Correlation	−0.313 **	0.148	1	0.196 *
	Sig. (2-tailed)	00.001	0.133		0.046
Hamilton D scale	Pearson Correlation	−0.039	−0.104	0.196 *	1
	Sig. (2-tailed)	0.694	0.292	0.046	

* *p* < 0.05, ** *p* < 0.01.

**Table 3 biomedicines-12-02646-t003:** Correlation between the Hamilton scale, the PANSS scale and its subscales, and the BPRS scale.

		Hamilton D	PANSS	PANSS Positive Subscale	PANSS Negative Subscale	PANSS Disorganized Subscale	BPRS
Hamilton D	Pearson Correlation	1	0.275 **	0.240 *	0.156	0.295 **	0.383 **
	Sig. (2-tailed)		0.005	0.014	0.113	0.002	0.000
PANSS	Pearson Correlation	0.275 **	1	0.835 **	0.806 **	0.954 **	0.911 **
	Sig. (2-tailed)	0.005		0.000	0.000	0.000	0.000
PANSS positive subscale	Pearson Correlation	0.240 *	0.835 **	1	0.493 **	0.738 **	0.785 **
	Sig. (2-tailed)	0.014	0.000		0.000	0.000	0.000
PANSS negative subscale	Pearson Correlation	0.156	0.806 **	0.493 **	1	0.730 **	0.719 **
	Sig. (2-tailed)	0.113	0.000	0.000		0.000	0.000
PANSS disorganized subscale	Pearson Correlation	0.295 **	0.954 **	0.738 **	0.730 **	1	0.888 **
	Sig. (2-tailed)	0.002	0.000	0.000	0.000		0.000
BPRS	Pearson Correlation	0.383 **	0.911 **	0.785 **	0.719 **	0.888 **	1
	Sig. (2-tailed)	0.000	0.000	0.000	0.000	0.000	

* *p* < 0.05, ** *p* < 0.01.

**Table 4 biomedicines-12-02646-t004:** Correlation between Hamilton depression rating scale and dissociation scale, fixation scale, Hamilton anxiety rating scale, and obsessive-compulsive symptoms rating scale.

		Hamilton D Scale	Dissociation Scale	Fixation Scale	Hamilton A Scale	OCS Scale
Hamilton D	Pearson Correlation	1	0.133	−0.139	0.719 **	0.256 **
	Sig. (2-tailed)		0.175	0.158	0.000	0.009
Dissociation scale	Pearson Correlation	0.133	1	−0.467 **	0.209 *	0.335 **
	Sig. (2-tailed)	0.175		0.000	0.032	0.000
Fixation	Pearson Correlation	−0.139	−0.467 **	1	−0.270 **	−0.121
	Sig. (2-tailed)	0.158	0.000		0.005	0.218
Hamilton anxiety scale	Pearson Correlation	0.719 **	0.209 *	−0.270 **	1	0.208 *
	Sig. (2-tailed)	0.000	0.032	0.005		0.033
OCS scale	Pearson Correlation	0.256 **	0.335 **	−0.121	0.208 *	1
	Sig. (2-tailed)	0.009	0.000	0.218	0.033	

* *p* < 0.05, ** *p* < 0.01.

**Table 5 biomedicines-12-02646-t005:** Results of the regression analysis. The factors with a statistically significant influence on the occurrence of depressive symptoms and their coefficients are presented.

	R2	β	t	p (sig)
Step 1 PANSS disorganized	0.295	0.148	3.136	0.002
Step 2 Duration of untreated symptoms	0.351	0.174	3.611	0.044
Step 3 OCS scale	0.409	0.150	3.098	0.022

**Table 6 biomedicines-12-02646-t006:** Statistical significance of established models when conducting regression analysis.

Model		Unstandardized Coefficients		Standardized Coefficients	t	Sig.
		B	Std. Error	Beta		
1	(Constant)	7.126	1.568		4.544	0.000
	PANSS disorganized	0.148	0.047	0.295	3.136	0.002
2	(Constant)	7.576	1.560		4.855	0.000
	PANSS disorganized	0.174	0.048	0.347	3.611	0.000
	Duration of untreated psychosis	−0.086	0.042	−0.196	−2.040	0.044
3	(Constant)	6.513	1.594		4.085	0.000
	PANS disorganized	0.150	0.048	0.299	3.098	0.003
	Duration of untreated psychosis	−0.098	0.042	−0.223	−2.351	0.021
	OCS scale	0.111	0.048	0.220	2.329	0.022

## Data Availability

The raw data supporting the conclusions of this article will be made available by the authors upon reasonable request.

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
