# Peer review of "Untangling Depression in Schizophrenia: The Role of Disorganized and Obsessive-Compulsive Symptoms and the Duration of Untreated Psychosis"

_biomedicines, 2024, doi:10.3390/biomedicines12112646_

Round 1

Reviewer 1 Report

Comments and Suggestions for Authors

The manuscript presented by Panov et al. explores the influence of disorganized, obsessive-compulsive symptoms and duration of untreated psychosis as factors related to depression in schizophrenic patients. The manuscript's main idea is interesting and has scientific value but I have some concerns that must be addressed before publication.

1) The title is too big. The title must have only one sentence.

2) The manuscript format must be corrected. There are different types of letters.

3) "All these deviations have an impact on the functioning of the immune system and the lipid profile /Tanaka M at al 2022; Correia BSB et al 2021/. " This sentence is "lost" in the text. 

4) Please translate "3.Резултати" to English.

5) Table 1. In which unit of measurement is the height?

6) Table 1. Regarding sex, What does mean 20/25?

7) the results presented in Tables 2-6 must be explored in the text.

Comments on the Quality of English Language

Minor editing of English language required.

Author Response

Thank you very much for your recommendations

  • The title is too big. The title must have only one sentence.

Thanks for the recommendation. the title has been changed to make it more concise and clear

  • The manuscript format must be corrected. There are different types of letters.

The format of the article has been corrected and the letters also=

  • "All these deviations have an impact on the functioning of the immune system and the lipid profile /Tanaka M at al 2022; Correia BSB et al 2021/. " This sentence is "lost" in the text. 

Thank you for the remark. The sentence has been changed. The idea is to present that the schizophrenia process is not just mental disturbance but the general condition affecting the whole the body and the brain.

  • Please translate "3.Резултати" to English.

Sorry for the omission, a change has been made

  • Table 1. In which unit of measurement is the height?

Thanks for the clarification - in centimeters. It has been changed in the text.

  • Table 1. Regarding sex, What does mean 20/25?

Thank you for your remark and sorry for my mistake. The attitude of men to women and actually is 39/66

  • The results presented in Tables 2-6 must be explored in the text.

Explanations of the tables are made in the text. Thanks for the recommendation. These explanations were indeed lacking, and without them it is more difficult to explain the results.

The author

Reviewer 2 Report

Comments and Suggestions for Authors

While reviewing the manuscript “Factors associated with depression in patients with schizophrenia. Influence of disorganized, obsessive-compulsive symptoms and duration of untreated psychosis. Do they give us a starting point for assessing which condition is basic and primary?” The conclusions of the study point out that although schizophrenia patients often have persistent depressive symptoms, there is some biological confrontation between psychotic symptoms and depressive symptoms, however, I would like to mention several issues, and I think it should be solved before I suggest publishing this manuscript.

Comment 1: Please explain how epigenetic factors influence the development of schizophrenia and antipsychotic treatment effects.

Comment 2: Please explain why depressive symptoms have different prognoses in different schizophrenia patients.

Comment 3: Whether sociodemographic factors should be regarded as independent predictors of depressive symptoms or whether they are merely consequences of the persistence of the main symptoms of schizophrenia.

Comment 4: Whether exclusion of sociodemographic data impacts the comprehensiveness and accuracy of study results when analyzing factors associated with depressive symptoms.

Comment 5: Whether the interaction of depressive symptoms and psychotic symptoms in treatment suggests the need for a new treatment to cope with both symptoms.

Comments on the Quality of English Language

Minor editing of English language required.

Author Response

Dear Reviewer

Thank you very much for your questions and comments

Comment 1: Please explain how epigenetic factors influence the development of schizophrenia and antipsychotic treatment effects.

Epigenetic factors play a significant role in the development of schizophrenia by influencing gene expression. Environmental factors such as stress, trauma, or polution can trigger epigenetic modifications of the DNA, such as DNA methylation or histone modification, which may affect neurotransmitter systems implicated in schizophrenia. These changes can alter brain function and thus contribute to the onset and progression of the psychosis. In terms of treatment, epigenetic modifications may influence an individual’s response to antipsychotic medications. Antipsychotic medications themselves are also epigenetic factors. On the other hand certain epigenetic changes may lead to altered receptor sensitivity or drug metabolism, impacting the effectiveness of antipsychotic medications and potentially contributing to drug resistance or side effects. Thank you for your question.

Comment 2: Please explain why depressive symptoms have different prognoses in different schizophrenia patients.

          The prognosis of depressive symptoms in schizophrenia varies and depend on several factors, including the type of schizophrenia, comorbid conditions, the duration of untreated psychosis, and the presence of specific symptoms such as disorganization or obsessive-compulsive traits. Some individuals may experience persistent depressive symptoms as part of the chronic course of the illness, while others may have intermittent or situational depressive episodes. Additionally, the response to antipsychotic treatment and its side effects can influence depressive symptomatology. The heterogeneous nature of schizophrenia itself, combined with individual variability in neurobiology and psychosocial factors, contributes to the differing prognoses of depression in schizophrenia patients.

Comment 3: Whether sociodemographic factors should be regarded as independent predictors of depressive symptoms or whether they are merely consequences of the persistence of the main symptoms of schizophrenia.

          Sociodemographic factors, such as age, gender, socioeconomic status, and educational level, can be both independent predictors of depressive symptoms and consequences of the persistence of the primary symptoms of schizophrenia. For instance, lower socioeconomic status and lack of social support may contribute to the onset or exacerbation of depressive symptoms by increasing stress levels or reducing access to treatment. At the same time, individuals with chronic or more severe schizophrenia symptoms may face difficulties in maintaining stable employment, relationships, and social networks, leading to sociodemographic challenges. Therefore, it is important to view sociodemographic factors as both influencing the development of depressive symptoms and as consequences of the illness, as they often interact in a bidirectional manner.

          We can and should consider socio-economic factors as independent predictors of the appearance of psychotic symptoms, but in such a case we should analyze patients with the first or, as a last resort, in the initial stages of schizophrenia. This does not apply to our patients, who have consecutive psychotic episodes, such as one of them with a long duration of the schizophrenic process /as seen in table 1 with an average duration of 11.77 years/ so in them these factors are a consequence of the long duration of the schizophrenic process. Even if in someone with a shorter duration they play a role in the onset of the disease, in others they will be a consequence of its progression. In this case, we will not compare analogous situations in these patients. On the other hand we have overall global incidence of schizophrenia that remains relatively consistent, typically estimated at about 1% of the population and also not to forget so called Social Drift Hypothesis whiсh can explain social situation of the patients with schizophrenia.

Hollingshead, A. B., & Redlich, F. C. (1958). Social class and mental illness: A community study. Wiley.ocioeconomic consequences.

Veling, W., Susser, E. S., & Hoek, H. W. (2008). The epidemiology of schizophrenia and social factors. Schizophrenia Bulletin, 34(4), 738–746. https://doi.org/10.1093/schbul/sbn052

Comment 4: Whether exclusion of sociodemographic data impacts the comprehensiveness and accuracy of study results when analyzing factors associated with depressive symptoms.

          We think the answer to the above question also contains part of the answer to this question. In the present study, we tried to analyze the purely clinical factors that could relatively easily be analyzed in a clinical setting by looking for answers about the intertwined psychopathological phenomena and the interrelationships between them. We are trying to understand your question in the context that it is difficult to separate the patient from the socioeconomic factors that surround him. With the development of the schizophrenia process, almost all patients fall to similar socioeconomic conditions, i.e. even those who lived in a rich socio-economic environment / the phenomenon described above Social Drift Hypothesis /. Even in such a socioeconomic environment, some develop depressive complaints to the point of clinical manifestation, others do not. This was our train of thought when we undertook this observation and analysis. Thanks for your question.

Comment 5: Whether the interaction of depressive symptoms and psychotic symptoms in treatment suggests the need for a new treatment to cope with both symptoms.

The link between depressive and psychotic symptoms in schizophrenia suggests that current treatments might need to be adjusted or new ones developed to better address both types of symptoms. Antipsychotic medications mainly treat psychotic symptoms like delusions and hallucinations but may not be effective for depression, and in some cases, they might make it worse. On the other hand, antidepressants can help with depression but don't typically treat the psychotic symptoms. A combined approach using both antipsychotics and antidepressants, or other new treatments, may be needed to effectively manage both depressive and psychotic symptoms. Research into treatments that target both symptom types at once could improve outcomes for patients and reduce the impact of the illness. There are some new medications that give hope and demand for wider effectiveness in relation to individual psychopathological phenomena.

Thank you

The authors

Reviewer 3 Report

Comments and Suggestions for Authors

The authors conducted a preliminary study of factors related to and influencing the appearance of depressive symptoms in patients with schizophrenia

Major comments:

Authors should clearly state the purpose of their research in the introduction. 

Authors should clearly state the novelty of their research.

The small sample size, unfortunately, does not allow for reliable conclusions to be drawn. The authors appear to be studying a subgroup of schizophrenia patients who also have symptoms of obsessive-compulsive disorder. The term for this type of patients is known in the literature - schizoobsessive disorder. Consequently, the findings of this paper can be applied only to this subgroup of schizophrenia patients.

To study factor interactions in such a complex condition, in addition to the small sample size, the authors lack other control groups to identify reliable interactions between the factors under study, e.g., a group of patients with depression and obsessive-compulsive disorder but without symptoms of schizophrenia, a group of relatively healthy individuals with depression only without symptoms of OCD, schizophrenia and other related conditions. 

Minor comments:

References in the text should be formatted according to the journal's rules.

The title of section 3 needs to be translated into English.

Comments on the Quality of English Language

Overall the English is good, may need a little proofreading

Author Response

Dear reviewer

Thank you very much for your valuable comments and advices.

Authors should clearly state the purpose of their research in the introduction. 

Changes have been made to the introduction section

Authors should clearly state the novelty of their research.

The purpose and objectives of the study are stated at the end of the introduction section in a separate paragraph

The small sample size, unfortunately, does not allow for reliable conclusions to be drawn. The authors appear to be studying a subgroup of schizophrenia patients who also have symptoms of obsessive-compulsive disorder. The term for this type of patients is known in the literature - schizoobsessive disorder. Consequently, the findings of this paper can be applied only to this subgroup of schizophrenia patients.

Thank you very much for your comment. We do not analyze patients with comorbid obsessive-compulsive disorder /respectively schizoobsessive one/ we use questionnaire for the analysis of obsessive-compulsive symptoms, such as the DOCS scale. So we analyzed obsessive-compulsive symptoms in patients with schizophrenia.

To study factor interactions in such a complex condition, in addition to the small sample size, the authors lack other control groups to identify reliable interactions between the factors under study, e.g., a group of patients with depression and obsessive-compulsive disorder but without symptoms of schizophrenia, a group of relatively healthy individuals with depression only without symptoms of OCD, schizophrenia and other related conditions. 

In the section limitations of our study, we have described the weaknesses of our study. Thanks for the remark. We will take them into account in future studies.

The title of section 3 needs to be translated into English.

Thank you. The translation is done.

The author

Round 2

Reviewer 3 Report

Comments and Suggestions for Authors

The authors responded to key comments and correctly pointed out the shortcomings of their work. I have no further comments and propose to accept the manuscript for publication.